# Mechanistic Insights into the Release of Doxorubicin from Graphene Oxide in Cancer Cells

**DOI:** 10.3390/nano10081482

**Published:** 2020-07-29

**Authors:** Erica Quagliarini, Riccardo Di Santo, Daniela Pozzi, Paolo Tentori, Francesco Cardarelli, Giulio Caracciolo

**Affiliations:** 1Department of Chemistry, Sapienza University of Rome, P.le A. Moro 5, 00185 Rome, Italy; erica.quagliarini@uniroma1.it; 2Department of Molecular Medicine, Sapienza University of Rome, Viale Regina Elena 291, 00161 Rome, Italy; riccardo.disanto@uniroma1.it; 3Center for Nanotechnology Innovation@NEST (CNI@NEST), Istituto Italiano di Tecnologia, Piazza San Silvestro 12, 56127 Pisa, Italy; paolo.tentori@sns.it; 4NEST Laboratory, Scuola Normale Superiore, Piazza San Silvestro 12, 56127 Pisa, Italy; francesco.cardarelli@sns.it

**Keywords:** doxorubicin, functional materials, graphene oxide, nanochemotherapeutics

## Abstract

Liposomal doxorubicin (L-DOX) is a popular drug formulation for the treatment of several cancer types (e.g., recurrent ovarian cancer, metastatic breast cancer, multiple myeloma, etc.), but poor nuclear internalization has hampered its clinical applicability so far. Therefore, novel drug-delivery nanosystems are actively researched in cancer chemotherapy. Here we demonstrate that DOX-loaded graphene oxide (GO), GO-DOX, exhibits much higher anticancer efficacy as compared to its L-DOX counterpart if administered to cellular models of breast cancer. Then, by a combination of live-cell confocal imaging and fluorescence lifetime imaging microscopy (FLIM), we suggest that GO-DOX may realize its superior performances by inducing massive intracellular DOX release (and its subsequent nuclear accumulation) upon binding to the cell plasma membrane. Reported results lay the foundation for future exploitation of these new adducts as high-performance nanochemotherapeutic agents.

## 1. Introduction

The anthracycline doxorubicin (DOX) is a common first-line therapy for numerous human pathological conditions such as breast cancer, ovarian cancer, multiple myeloma and Kaposi’s sarcoma. In spite of this, DOX recognized adverse side effects, above all cardiotoxicity [1] and nephrotoxicity [2], encouraged the development of alternative nanocarrier-based strategies for safer systemic delivery of DOX. The aim was, in particular, at exploiting the nanocarrier intrinsic ability to protect the drug from degradation, prolong its circulation lifetime, reduce drug dose, finally having a positive impact on drug biodistribution, pharmacokinetics and therapeutic efficacy [3]. In this regard, liposomal DOX (L-DOX) in all its variants (e.g., Doxil^®^, Doxoves^®^ and LipoDox^®^) is a paradigmatic example. It has been the first liposomal drug approved by the Food and Drug Administration (FDA) and remains, at present, the drug of choice for the treatment of a number of cancer-related pathologies (mentioned above) [4]. Despite the several advantages conferred by L-Dox formulations (i.e., increased circulation lifetimes as well as enhanced antitumor activity and less toxicity in respect to the free drug), their clinical application is far from being established [5], i.e., almost 10 years after L-DOX-related patents expired, there is still no FDA-approved “generic L-DOX” available. General consensus has been reached on the crystalline form of DOX within the lipid vesicles being a major rate-limiting factor for L-DOX efficacy [4]. On one side, it has been suggested that crystalline DOX may favor the rupture of lipid vesicles with the consequent extracellular release of a fraction of the transported payload [6]. On the other side, the largest fraction of internalized crystallized drug stays trapped within liposomes and is shuttled to lysosomes for degradation [7]. Lastly, its interaction with nuclear DNA could be disadvantaged compared to that of its monomeric counterpart.

Based on all these considerations, it is clear that novel nanocarriers that are capable of overcoming the limitations of standard L-DOX formulations are highly desirable in the field. Graphene oxide (GO) is considered a promising nanocarrier due to its peculiar physical–chemical properties such as high surface area, high water dispersibility and good biocompatibility [8]. Compared to other nanocarriers, GO exhibits favorable drug loading capacity due to the two-dimensional planar arrangement of carbon atoms that allows a total exposure of functional groups to the chemical surrounding [9]. Moreover, epoxydic and carboxylic groups on the GO surface can be used to selectively bind drug molecules, thus enhancing drug solubility in aqueous media [10,11,12]. On the other side, lateral dimensions of GO nanosheets have limitations regarding blood–brain transport, renal clearance, biodegradation and toxicity that have substantially impaired its clinical application so far [13,14].

Here, we explore the potentiality of GO for the efficient loading and intracellular release of DOX in MCF-7 and MDA-MB 231 breast cancer cells [15]. By standard confocal imaging, exploiting DOX intrinsic fluorescence, we demonstrate that GO promotes massive intracellular delivery and subsequent nuclear accumulation of DOX molecules and that such an outcome is followed by high anticancer activity. By phasor-fluorescence lifetime imaging microscopy (FLIM) analysis, in turn, we observe that the DOX molecules within the cytoplasm and nucleus of cells treated with GO-DOX have a characteristic lifetime signature similar to that found in cells treated with the free drug. In addition, GO is detected as attached/adsorbed to the cell plasma membrane, mostly in its unloaded configuration. These observations support the hypothesis of effective drug release from the GO-DOX adduct upon binding of this latter to the cell plasma membrane. Results reported here pave the way to future applications of GO in drug delivery research.

## 2. Materials and Methods

### 2.1. Preparation of GO-DOX Complexes

Graphene oxide (GO) water dispersion was purchased from Graphenea (San Sebastián, Spain). Doxorubicin hydrochloride (DOX) was acquired from Formumax Scientific (Sunnyvale, CA, USA). GO-DOX complexes were prepared by mixing 1 mL of GO (0.4 mg/mL) with 1 mL of DOX water dispersion, at various concentrations. Doing so, we obtained GO-DOX complexes at different GO/DOX weight ratios (R_W_), i.e., R_W_ = 3.00, 2.00, 1, 0.50 and 0.33. The mixtures were adjusted with 0.01 M NaOH to have a final pH value of 8 and sonicated for 3 min at 125 W with vibra cell sonicator VC505 (Sonics and Materials, Suffolk, UK). The resulting solutions were left in darkness overnight at 37 °C. Finally, to remove the unbounded DOX, the mixtures were subjected to centrifugation (Hermle Z 216 MK, Hermle Labortechnik, Germany) at 18,620 RCF for 30 min at 4 °C and the pellet was resuspended with ultrapure water. This procedure was repeated three times. After each washing step, the supernatant was collected and unloaded DOX was quantified by UV-VIS experiments.

### 2.2. UV-VIS Experiments

UV-VIS experiments were performed using a UV-VIS spectrophotometer (JASCO V-570, JASCO Corporation, Tokyo, Japan). As a first step, a DOX calibration curve was generated from a series of DOX solutions with different concentrations (Appendix A). The calibration curve was obtained by plotting the absorbance 480 nm versus DOX concentration and was treated by linear regression analysis. The linearity of the response of the drug was verified at 1–20 μg/mL concentrations. Thus, the DOX loading efficiency (Equation (1)) and loading capacity (Equation (2)) were calculated by the following equations [16]:(1)$$\mathrm{Loading}\;\mathrm{Efficiency}\;\left(\%\right)=\frac{\mathrm{Amount}\;\mathrm{of}\;\mathrm{total}\;\mathrm{DOX}-\mathrm{Amount}\;\mathrm{of}\;\mathrm{unloaded}\;\mathrm{DOX}}{\mathrm{Amount}\;\mathrm{of}\;\mathrm{total}\;\mathrm{DOX}}\times 100$$
(2)$$\mathrm{Loading}\;\mathrm{Capacity}\;\left(\frac{\mu g}{\mu g}\right)=\frac{\mathrm{Amount}\;\mathrm{of}\;\mathrm{total}\;\mathrm{DOX}-\mathrm{Amount}\;\mathrm{of}\;\mathrm{unloaded}\;\mathrm{DOX}}{\mathrm{Amount}\;\mathrm{of}\;\mathrm{GO}}$$

### 2.3. Dynamic Light Scattering

Size of GO and GO-DOX complexes was determined by dynamic light scattering (DLS). DLS experiments were performed by means of a Zetasizer Nano ZS (Malvern Panalytical, Malvern, UK) at 25 °C. To this end, 100 μL of GO or GO-DOX complexes were diluted with 900 μL of distilled water (final volume = 1 mL). The results are given as mean ± standard deviation of three independent replicates.

### 2.4. Cell Culture

Breast cancer cell lines (MDA-MB 231 and MCF-7 cells) were purchased from ATCC (Manassas, VA, USA). Cancer cells were preserved in Dulbecco’s modified Eagle medium (DMEM) supplemented with 10% fetal bovine serum (FBS), 1% penicillin and streptomycin and 1% of L-glutamine. Cells were maintained at 37 °C and 5% CO_2_ atmosphere.

### 2.5. Cell Metabolic Activity Experiments

For cell metabolic activity experiments, Doxoves^®^ and GO-DOX complexes were administrated to MCF-7 and MDA-MB 231 breast cancer cells. In accordance with previous literature [6,17], two DOX concentrations (13 µg/well; 3 µg/well) were tested. Cell metabolic activity of MCF-7 and MDA-MB 231 cells was evaluated by 3-(4,5-dymethyl thiazol 2-y1)-2,5-diphenyl tetrazolium bromide assay (MTT assay, mitochondrial respiration analysis, Sigma-Aldrich, Milan, Italy), according to Mosmann protocol [18]. Water-soluble tetrazolium salt (WST-8) or Cell Counting Kit 8 (MTS) may be used as reliable alternative assays [19]. Briefly, cells were seeded on 96-well plates (10,000 cells/well) for 24 h. After 3 h incubation in Optimem, the medium was replaced with DMEM 10% FBS, cells were incubated for 48 h at 37 °C, and 100 µL of isopropyl alcohol was added to each well to dissolve the formazan salt. Finally, the absorbance of each well was obtained with Glomax Discover System (Promega, Madison, WI, USA).

### 2.6. Confocal Imaging and Phasor-FLIM Experiments

Each imaging experiment started 3 h after the incubation of the cells with GO-DOX and Doxoves^®^, without washing the cells to remove the excess of non-internalized compounds. For imaging, a Leica SP5 confocal microscope (Leica Microsystems AG, Wetzlar, Germany) was used, equipped with an Argon laser for excitation at 488 nm. Doxorubicin emission was collected in the 500-650 nm range. All experiments were performed by using the same imaging settings (e.g., magnification, laser power, etc.). The quantification of the recorded signal was carried out by Image J software (National Institutes of Health, Bethesda, Maryland, USA). During the experiment, the cells were maintained in a thermostatic chamber (37 °C, 5% CO_2_). For FLIM experiments, all analyzed samples were excited at 470 nm with a pulsed diode laser operating at 40 MHz (average power: 10-20 μW at the sample) and collecting the emission in the 500-650 nm range by a photomultiplier tube interfaced with a time-correlated single photon counting card and setup (PicoHarp 300, PicoQuant, Berlin). Phasor analysis of lifetime data was performed by SimFCS software (www.lfd.uci.edu, University of California at Irvine).

### 2.7. Statistical Analysis

Three replicates of each sample were performed for both cell lines. Values were expressed as average ± standard deviation. Statistical significance was evaluated using a Student’s *t*-test (* *p* < 0.05; ** *p* < 0.01).

## 3. Results and Discussion

Previous studies explored factors affecting DOX loading on GO sheets [20]. Among them, pH adjustment is considered as a fundamental step for efficient DOX loading by non-covalent bonding. However, controversial results about pH optimization have been reported. While some studies indicated that DOX loading on GO is maximum at neutral pH [21], some other identified basic pH as optimal for loading [12,22]. In a series of preliminary experiments (Appendix A), we found clear evidence that in a basic environment (pH = 8) the complexes show a lower aggregation state respect to other pH conditions, thus higher DOX loading on GO nanosheets can be achieved. Following pH adjustment, the role of the GO/DOX weight ratio (R_W_) on DOX loading was explored by UV-Vis experiments. To this end, GO-DOX complexes were centrifuged, the pellet was washed three times with ultrapure water and the supernatants carefully collected (Figure 1a).

Figure 1b shows the absorbance spectra measured from wash supernatants in the wavelength range of 350–650 nm. After the third washing step, the absorption peak of DOX was indistinguishable from the background thus ensuring that no significant amount of unbound DOX was present in the sample. The pellet was finally resuspended in ultrapure water (Figure 1a, “P”) and the measured peak absorbance (Figure 1b, “P”) was used for quantification of DOX loading efficiency and capacity (Figure 1c). As Figure 1c clearly shows, R_W_ = 0.33 allowed one to maximize the DOX loading capacity and was used in the following experiments. The average hydrodynamic size of GO and GO-DOX complexes in aqueous solution was measured by DLS as previously done for large size GO sheets [23]. Data reported in Appendix A showed that GO-DOX complexes and pristine GO were similar in size with a hydrodynamic diameter (DH) distribution peaked around at 700 nm. Next, GO-DOX complexes were administered to MCF-7 and MDA-MB 231 breast cancer cell lines at two DOX concentrations (13 µg/well and 3 µg/well). Pristine GO (i.e., not loaded with DOX) and clinically approved L-DOX in the form of Doxoves^®^ were used as controls. Cell metabolic activity was determined using the MTT assay (Figure 2).

Cell metabolic activity of MCF-7 and MDA-MB 231 breast cancer cells treated with pristine GO stayed close to 100% thus confirming that cell death was exclusively due to DNA damage produced by DOX. The administration of free DOX led to a marked reduction of metabolic activity in both cell lines. This result is in line with several previous investigations showing that DOX is one of the most powerful chemotherapeutic agents developed so far. However, as free drugs often undergo premature degradation in the biological environment, encapsulation within nanocarriers is essential to enhance their bioavailability and boost cellular uptake. GO, despite the above-mentioned limitations, is a promising nanoplatform for the delivery of DOX. At high drug concentration, anticancer efficacy of GO-DOX complexes was much higher (residual cell metabolic activity at 9%, 24 h after treatment) than that of commercial Doxoves^®^ (63%). Of note, similar results were obtained with MDA-MB 231 cells (cell metabolic activity 13% and 79% following treatment with GO-DOX complexes and Doxoves^®^ respectively) and, in both cell lines, at low drug concentration (Figure 2).

Many anticancer drugs, including DOX, act by intercalating nuclear DNA [24]. Thus, to fully exploit its anticancer efficacy, DOX must be efficiently internalized within cancer cells and subsequently delivered to the cell nucleus. We, therefore, used confocal imaging to explore the intracellular distribution of DOX in cancer cells. Figure 3 (panels a and b) shows representative confocal images of MCF-7 cells treated with Doxoves^®^ and GO-DOX complexes.

It is clear, already from visual inspection of images, that GO is able to promote higher cell internalization of DOX as compared to Doxoves^®^. In particular, when MCF-7 cells are treated with GO-DOX, cell nuclei exhibit higher fluorescence signals. By contrast, Doxoves^®^ treatment produces minor, if any, nuclear accumulation of DOX molecules. The histogram plot in Figure 3c reports quantitatively on the nuclear and cytoplasmic signals detected. Worthy of mention, the nuclear fluorescence in cells treated with GO-DOX complexes is five-fold higher as compared to that detected in cells treated with Doxoves^®^. Results were confirmed in MDA-MB 231 cells (Figure 3d–f). Overall, standard confocal imaging suggests that the marked reduction in cell metabolic activity produced by GO-DOX complexes may reflect the higher nuclear content in DOX molecules. Intracellular (and intranuclear) DOX molecules, in turn, are supposed to be free from the GO carrier to explain their functional effect on cell metabolic activity. To support this latter conjecture, we performed fluorescence lifetime imaging microscopy (FLIM) experiments on GO-DOX-treated cells, using the free drug as control (Figure 4). Among the properties of fluorescence, fluorescence lifetime, in fact, can be revelatory of the nanoscale supramolecular organization of the emitter molecule (in this case DOX). In fact, it is expected that DOX in its free form in solution can be distinguished from its counterpart adsorbed/attached to GO or to biological membranes based on the measured characteristic lifetime. The phasor approach to FLIM data is used here as a fit-free, fast and graphical method to extract the quantitative information encrypted in time-domain lifetime measurements [25]. First, we probed the phasor-FLIM signature of free DOX and GO-DOX in solution (Appendix A). Compared to free DOX (mono-exponential lifetime at 1 ns, in keeping with previous reports [26]), GO-DOX shows an average decrease in the characteristic lifetime, which becomes slightly multi-exponential in nature (Appendix A). This latter effect can be ascribed to the contribution of GO intrinsic fluorescence lifetime (measured here and found to be 0.1 ns, Appendix A) to the average lifetime of the complex. Worthy of mention, the only minor effect on DOX lifetime induced by its adsorption on GO suggests that that the drug can explore, dynamically, multiple conformations in the complex with GO (in agreement with previous reports [27]).

At this point, we treated MDA-MB 231 cells with DOX and, in parallel, with GO-DOX. The phasor-FLIM analysis is reported in Figure 4.

The fluorescence lifetime spectra measured in each pixel of the image are mapped onto the “phasor” plot: this is made up out of two numbers: the real and imaginary parts (‘g’ and ‘s’, amplitude and phase, respectively) of the first harmonic of the Fourier transform of the fluorescence lifetime. Thus, the phasor plot contains clouds of points that correspond to pixels with similar lifetime spectra. These clouds can be selected by specific ROIs, as indicated in Figure 4a,c. As somewhat expected, the characteristic lifetime signature of free DOX transforms into a broad distribution of multi-component lifetimes upon interaction (and mixing) with the complex ensemble of extracellular and intracellular autofluorescent species, which are multi-exponential in nature [28]. In particular, two subclusters are easily recognizable in the phasor plot, one corresponding to DOX molecules localized in the nucleus (red circle in Figure 4a, red pixels in Figure 4b), another corresponding to DOX molecules localized in the cytoplasm (green circle in Figure 4a, green pixels in Figure 4b). Please note, also, that the lifetime of DOX in the nucleus is close to that of free DOX in water (indicated by the red point in Figure 4a,c) while, by contrast, DOX lifetime in the cytoplasm shows a sensibly different distribution pattern that can be reasonably ascribed to interaction of DOX with biological membranes (DOX associated to surfaces, in fact, is known to have a lifetime similar to that extrapolated here from data and located at approximately 4 ns [29], green point in the phasor plot in Figure 4a). What is important for the purpose of this work, however, is that GO-DOX shows some relevant similarities with free DOX in terms of phasor-FLIM signatures within the cell. In particular, the same phasor-plot features described for free DOX are found in cells treated with GO-DOX: an elongated overall distribution with one subcluster corresponding to nuclear DOX (closer to DOX in water, highlighted in red) and another corresponding to cytoplasmic DOX (closer to DOX adsorbed to surfaces/membranes, highlighted in green; Figure 4c). Of note, the lifetime analysis unveils sites of putative GO-DOX attachment to the cell surface (and subsequent drug release), which are not visible in the intensity image (reported in Figure 4d, left panel). As highlighted in Figure 4c, the most important difference in the lifetime data from cells treated with GO-DOX is the presence of pixels with sensibly shorter lifetimes, in particular pointing towards the lifetime of nude GO (i.e., 0.1 ns, orange point). The triangle with vertices the lifetime of the three “pure” species (free DOX, DOX associated to membranes and nude GO) can represent a framework guiding data interpretation. As mentioned above, the phasor data lying along the line between free DOX and membrane-associated DOX recapitulates the main features observed treating cells with free DOX and can be interpreted as an indication that the drug is effectively released at the intracellular level. By contrast, data along the lines connecting nude GO with the other two species unveil the presence of pixels in the image where a variable amount of the carrier (GO) and the released drug (both free or associated to cellular membranes) is detected. We separated this population of pixels into subpopulations of a different color (see violet, orange and yellow cursors in Figure 4c). Of note, however, they all concur to highlight selected micrometric patches on the cell border (Figure 4d). We are prompted to ascribe these patches to sites of GO-DOX cell attachment and subsequent drug release. This observation is in agreement with recent findings obtained using graphene decorated with cyclodextrins and loaded with doxorubicin [30]. The authors, in fact, also due to lifetime analysis, evidenced the efficient cellular uptake of the whole adduct and the presence of DOX in the nucleus without the graphene carrier. Additionally, our data do not contradict previous evidences showing that GO likely binds to integrins at the plasma membrane of cancer cells, activates the integrin-FAK-Rho-ROCK pathway and makes cancer cells more susceptible to chemotherapeutic agents [31]. From a methodological point of view, our results corroborate the idea that FLIM can represent a quantitative platform to analyze DOX cellular uptake and release from nanocarriers, in line with a growing body of evidences from the literature (see for instance References [29,30,32,33]).

## 4. Conclusions

In summary, we have shown that GO can massively deliver free DOX in the nucleus of cancer cells by attaching to the cell membrane and releasing its payload into the cytosol. By this mechanism, GO leads to a significant increase in cell mortality with respect to approved L-DOX. We indeed see this proof of concept work as being part of a future workflow for all bionanoscience, nanomedicine and other areas in which shedding light on the mechanisms of GO-mediated drug delivery is fundamental to fully exploit its potential. The main limitation of the present study is undoubtedly the lateral size of GO-DOX complexes that could be too big for the systemic delivery of DOX [34]. Moreover, in view of in vivo applications, chemical modification of small size GO with hydrophilic polymers [35] such as polyethylene glycol (PEG) or polyethylene oxide (PEO) may also be necessary. Systemic administration of large size GO sheets would likely activate the mononuclear phagocytic system leading to fast removal from the bloodstream. On the other side, other topical medications such as intratumoral and subcutaneous location do not suffer limitations arising from the employment of large size delivery systems. Several biomaterials have been used for the local delivery of DOX to breast cancer, skin cancer and colorectal cancer [36,37]. Thus, we suggest that GO sheets used in this work may be useful for these applications. Likewise, clarifying other relevant aspects of the delivery process would be required to strengthen the significance of this study (e.g., apoptosis and invasion/migration) and will be the focus of future investigations. Among other perspectives, coating GO-DOX complexes with a biomolecular corona (i.e., the biomolecular layer surrounding nanomaterials in biological fluids [38,39,40]) to boost local delivery of DOX could be a promising strategy and will be also addressed in future studies.

## Figures and Tables

**Figure 1 nanomaterials-10-01482-f001:**
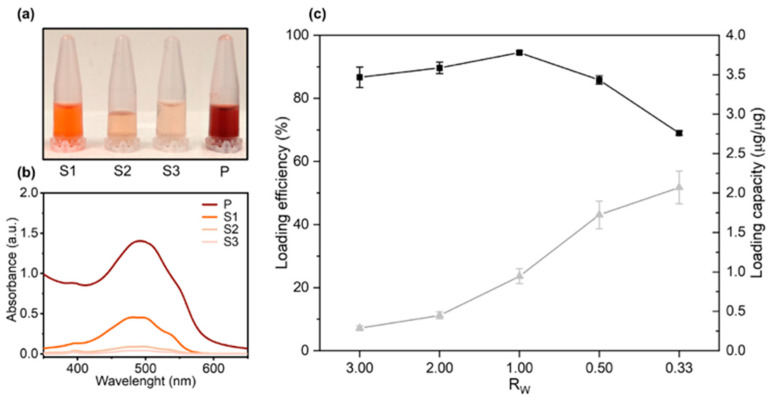
Loading efficiency and loading capacity of doxorubicin (DOX) on graphene oxide (GO) sheets as a function of the GO/DOX weight ratio R_W_. (**a**) GO-DOX complexes were centrifuged, and the pellet was washed three times with ultrapure water (indicated as S1, S2 and S3 respectively). (**b**) Absorbance spectra measured from the supernatants. After the third washing step, the pellet was resuspended in ultrapure water and the solution (P) used for (**c**) quantification of DOX loading efficiency (black squares) and capacity (grey triangles).

**Figure 2 nanomaterials-10-01482-f002:**
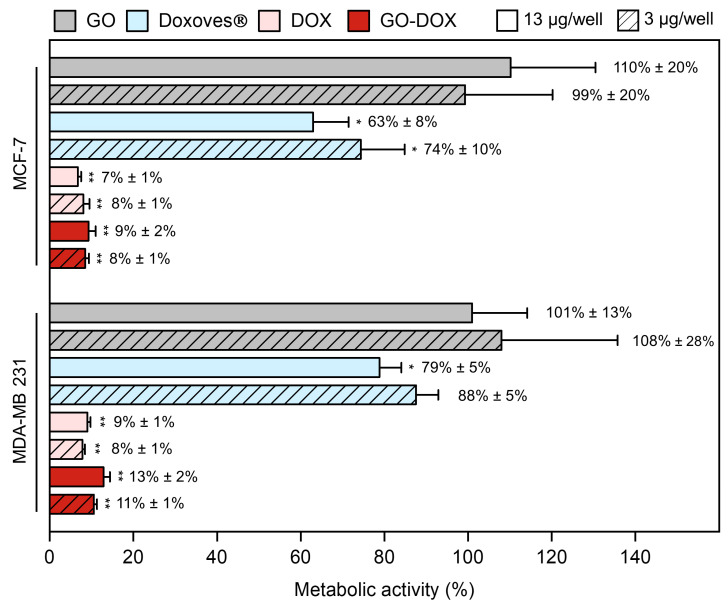
Cell metabolic activity of MCF-7 and MDA-MB 231 cells treated with graphene oxide (GO), commercial liposomal doxorubicin (Doxoves^®^), free doxorubicin (DOX) and DOX-loaded GO sheets (GO-DOX) expressed as a percentage with respect to untreated cells. Cell metabolic activity experiments were performed at two different DOX concentrations: i) 13 μg/well (empty bars) and 3 μg/well (diagonal pattern bars). Results are the average of three replicates ± standard deviation, normalized. Statistical significance was evaluated with respect to GO using a Student’s *t*-test (* *p* < 0.05; ** *p* < 0.01).

**Figure 3 nanomaterials-10-01482-f003:**
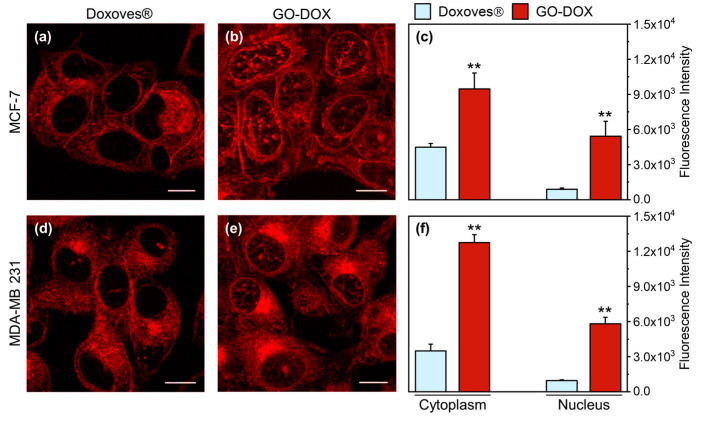
Representative confocal microscopy images of MCF-7 cells treated with (**a**) Doxoves^®^ and (**b**) doxorubicin (DOX)-loaded GO sheets GO-DOX complexes (GO/DOX weight ratio, Rw = 0.33). Experiments were performed at high DOX concentration (i.e., = 13 µg/well). (**c**) Fluorescence intensity analysis of the intracellular localization of DOX. (**d**–**f**) Analogously, confocal imaging and analysis were performed in MDA-MB 231 cells. Statistical significance was evaluated using a Student’s *t*-test (** *p* < 0.01) with respect to Doxoves^®^. Calculations reported in panels c and f were made using not less than 60 randomly selected cells. Scale bars are 10 µm.

**Figure 4 nanomaterials-10-01482-f004:**
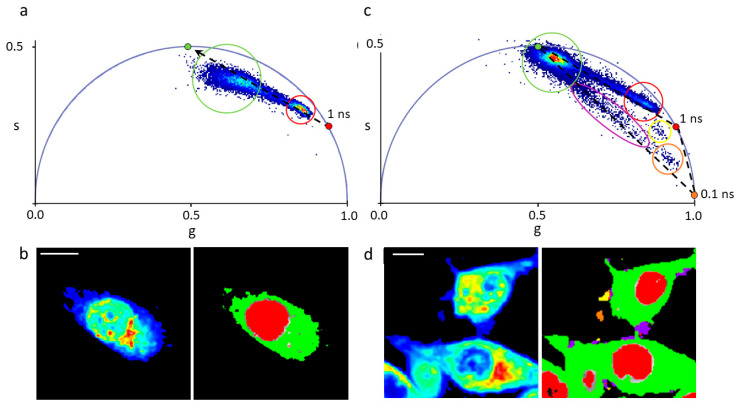
Phasor FLIM analysis of MDA-MB 231 cell exposed to DOX and GO-DOX. (**a**) Phasor representation of lifetimes measured in cells exposed to DOX. The phasor plot contains clouds of points that correspond to pixels with similar lifetime spectra. These clouds can be selected by specific regions of interest (ROIs). The green ROI, for instance, highlights the portion of pixels corresponding to DOX in the cytoplasm (see right panel in ‘b’, the cytoplasm is colored in green); the red ROI clearly identifies pixels corresponding to DOX in the nucleus (see right panel in ‘b’, the nucleus is colored in red); a dashed line is drawn across the phasor distribution and used to extrapolate the hypothetical position of a pure species on the universal circle (green point located at 4 ns) presumably corresponding to DOX associated to membranes. (**b**) Intensity (left) and lifetime (right) images of aMDA-MB 231 cell exposed to DOX (the same analyzed in (a)). The lifetime image is colored according to the ROIs in (a). (**c**,**d**) Same as before but for MDA-MB 2311 cells exposed to GO-DOX. Please note that the phasor-FLIM signature of cytoplasmic (green cursor in ‘c’, green pixels in ‘d’) and nuclear DOX signals (red cursor in ‘c’, red pixels in ‘d’) show clear similarity to that obtained for free DOX. By contrast, as expected, additional signals are present here, which are absent in cells treated with free DOX. In particular, as highlighted by the violet, orange and yellow cursors in ‘c’ (and pixels of the same color in ‘d’) there are micrometric patches, mostly associated with cell membranes, with variable amounts of the nude carrier (GO) and the released drug (both free or associated to cellular membranes). Scale bars: 10 µm.

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
