# Peer review of "Mechanistic Insights into the Release of Doxorubicin from Graphene Oxide in Cancer Cells"

_nanomaterials, 2020, doi:10.3390/nano10081482_

Round 1

Reviewer 1 Report

In this study, authors demonstrate that DOX-loaded Graphene Oxide exhibits lower cell viability compared to its L-DOX counterpart in human breast cancer cell lines in vitro. They also show that GO-DOX induces higher intracellular DOX release by using combination of live-cell confocal imaging and Fluorescence Lifetime Imaging Microscopy.

Comments:

  1. Although the studies are conducted well, they are preliminary. Only limited in vitro studies are performed.
  2. As stated in the conclusions, authors should have shown how GO leads to a significant increase in cell apoptosis with respect to approved L-DOX. Apoptosis experiments are missing. 
  3. pH is an important factor during preparation of GO-DOX complexes. Authors show that basic pH 8.0 is optimum for maximum DOX loading on GO. How will this be effected when the GO-DOX complex is added to the cell lines in vitro or given in vivo. Comment.

Reviewer 2 Report

This manuscript shows an interesting and easy approach to loading graphene oxide nanoparticles with doxorubicin, and then goes on to show enhanced nuclear uptake compared to liposomal dox for two breast cancer cell lines. The results demonstrate some of the potential for the drug delivery system, however there are a few concerns with the manuscript.

1) this is not a novel approach, as GO-dox has been explored for more than 10 years. Authors must better describe what the new contribution/approach is compared to literature.

2) Authors suggest Rw.0.33 is "best compromise." Why is this the best compromise? what if the user only wanted 1ug/ug or less for their treatment strategy... instead you present here an acceptable range for administering different desired doses

3) figure 2, its impossible to have more than 100% viability. the data you show here is metabolic activity, which indicates higher cell numbers (ie, cells are multiplying) or higher cell activity. Change the x axis label from viability to metabolic activity. What are the bars with the crossed lines on them, indicate in the legend? Also, what is the control here? untreated cells? what are these compared to? its not clear.

4) figure 3, this is not a cell viability assay. this is a nuclear dox localization assay. further, some key controls are not shown. 1)What is the fluorescence signal of GO-DOX dispersed in media without any cells present. 2) what is the background fluorescence of these cells without GO-DOX using the same microscope settings? those two images should be shown.

5) a live/dead or apoptosis (annexin V) assay would strengthen the study to quantify cell death.

6) a functional assay such as invasion/migration would greatly support the notion of GO-DOX as an effective dox delivery strategy.

7) grammar should be checked throughout the manuscript.

Reviewer 3 Report

This manuscripts describes the use of graphene oxide as a drug delivery vehicle for doxorubicin against two cellular models of breast cancer. A substantial improvement in intranuclear delivery of doxorubicin from graphene oxide sheets in comparison to commercially available L-Dox was shown. The manuscript is well written and the data on lifetime imaging was very convincing to describe the free dox release and intracellular distribution. I agree with the authors, while the nuclear payload delivery of Dox using GO is remarkable in these in vitro models, it will likely be limited in systemic delivery via intravenous injection. Despite PEGylation, these GO sheets (900 nm from DLS) will likely not circumvent the mononuclear phagocytic system and limit the tumour delivery and clinical efficacy of these particles.

Despite these limitations I would encourage the authors to position their work with a different scope. Could it not be possible to treat more accessible cancers other than breast? Such as skin cancer or even rectal/colon cancer? There has been work on localized chemotherapeutic delivery from patch applications applied directly over cancerous colon tissue, or even intratumoral injection This type of drug delivery system maybe better suited to such type of application. 

Since the data herein have shown promising in vitro results, the authors should consider testing this against other more accessible cancer cell models as highlighted above. Depending on the outcome of these data, this work would have significant impact in the field of drug delivery. For these reasons I am recommending a decision of acceptance following this major revision is completed with as compelling data on the models of breast cancer. Without these data, the impact of this work is solely limited to in vitro models. While the authors do suggest in vivo work is beyond the scope of this study, it is the next logical step and the authors do not communicate a compelling story for future work.

Please see additional comments below to further improve the quality of the manuscript.

Lns 46-50, the authors do highlight some limitations of L-Dox but have not recognized the substantial improvements in blood half-life or improvements in toxicity which were very important in the approval of this nanomedicine.

lns 143-144, why can't these data be reported in the supplement of this paper?

Please enhance the contrast and clarity of the cell fluorescence images in Figure 3, it is difficult to see peripheral intracellular boundaries.

Please refine the text of the caption for Figure 4, it is difficult to follow. Perhaps adding some labels to the figure itself may help.

Can the authors comment on why GO sheets were efficient at releasing free dox into the nuclei?

Round 2

Reviewer 1 Report

None

Author Response

No response will be given

Reviewer 2 Report

no more comments

Author Response

No response will be given